# Persistent *Mycobacterium tuberculosis* infection in mice requires PerM for successful cell division

**Ruojun Wang[1,2], Kaj Kreutzfeldt[1], Helene Botella[1†], Julien Vaubourgeix[1†], Dirk Schnappinger[1], Sabine Ehrt[1,2]***

[1]Department of Microbiology and Immunology, Weill Cornell Medical College, New York, United States; [2]Immunology and Microbial Pathogenesis Graduate Program, Weill Cornell Graduate School of Medical Sciences, Cornell University, New York, United States

**Abstract** The ability of *Mycobacterium tuberculosis* (Mtb) to persist in its host is central to the pathogenesis of tuberculosis, yet the underlying mechanisms remain incompletely defined. PerM, an integral membrane protein, is required for persistence of Mtb in mice. Here, we show that *perM* deletion caused a cell division defect specifically during the chronic phase of mouse infection, but did not affect Mtb's cell replication during acute infection. We further demonstrate that PerM is required for cell division in chronically infected mice and in vitro under host-relevant stresses because it is part of the mycobacterial divisome and stabilizes the essential divisome protein FtsB. These data highlight the importance of sustained cell division for Mtb persistence, define condition-specific requirements for cell division and reveal that survival of Mtb during chronic infection depends on a persistence divisome.

*For correspondence:
sae2004@med.cornell.edu

Present address: †MRC Centre for Molecular Bacteriology and Infection, Imperial College London, London, United Kingdom

Competing interests: The authors declare that no competing interests exist.

## Introduction

The World Health Organization reports that an estimated 23% of the world's population is latently infected with Mtb (https://www.who.int/tb/publications/global_report/en/). Although asymptomatic and non-contagious, latent infections progress to active disease in 5–15% of cases, causing clinical symptoms and allowing for disease transmission (*Pai et al., 2016*). The time before active disease develops following exposure to Mtb, ranges from a few months to two years (*Behr et al., 2018*). And, in regions with minimal TB transmission, Mtb can persist in individuals for decades before causing active disease (*Behr et al., 2018*). This remarkable ability of Mtb to persist inside its host contributes greatly to its success as a pathogen yet the mechanisms facilitating persistence remain incompletely defined. The term 'persistence' used in this report refers to the ability of bacteria to remain viable in the host for a prolonged period of time, and should be distinguished from 'persister cells', a subpopulation of transiently antibiotic-tolerant cells (*Fisher et al., 2017*).

The mouse model has been used to evaluate bacterial factors that allow Mtb to persist within its host. After aerosol infection, Mtb replicates logarithmically in mouse lungs until the onset of adaptive immunity arrests replication, leading to a chronic infection that is contained but not eliminated. Experiments with genetic Mtb mutants identified mycobacterial pathways that are crucial for persistence following the onset of adaptive immunity despite being dispensable for cell replication in the acute phase of infection. Among those pathways, metabolic adaption and modulation of the cell envelope play key roles in Mtb's persistence (*Ehrt et al., 2018*; *Glickman and Jacobs, 2001*).

We have previously reported that PerM, an integral membrane protein with 10 transmembrane helices is essential for Mtb persistence in the mouse model of infection. Disruption of *perM* (*rv0955*) by transposon insertion led to a minor growth defect in the acute phase of mouse infection, but a

strong attenuation in the chronic phase (*Goodsmith et al., 2015*). Further characterization of the mutant by transcriptome analysis, microscopy imaging and drug susceptibility assays suggested that PerM is associated with cell division. However, PerM's specific roles in cell division and how its absence resulted in a persistence defect in vivo remained unknown.

TB is a complex disease that is highly heterogeneous with regards to outcomes of infection, pathology and nature of bacterial populations (*Cadena et al., 2017*; *Logsdon and Aldridge, 2018*; *Rego et al., 2017*). In fact, the apparent static, non-replicating population that persists in chronically infected mice consists of a mixture of heterogeneous populations of replicating, non-replicating but alive, and dead bacteria (*Gill et al., 2009*). Using a 'replication clock', a plasmid-based reporter of bacterial replication, Gill et al. showed that Mtb replicates during the chronic phase of mouse infection, although at a much slower rate than in the acute phase. This discovery suggests that growth is not only important for increasing the bacterial burden during acute infection, it may also be required for Mtb to persist in chronically infected mice.

Here, we demonstrate that PerM is a mycobacterial divisome component and facilitates cell division by stabilizing FtsB, a conserved divisome protein known to be essential for the initiation of septation (*Buddelmeijer et al., 2002*; *Levin and Losick, 1994*; *Wu et al., 2018*). We document that although PerM is dispensable during the acute phase of mouse infection, it is required for Mtb cell division during chronic mouse infection and in host-relevant stress conditions, during which maintaining an optimal FtsB level becomes crucial. To our knowledge, PerM represents the first reported bacterial divisome component that is conditionally essential for cell division at different stages of mouse infection.

## Results

### Mtb requires PerM for optimal cell replication during chronic mouse infection

We previously reported that an Mtb *perM* transposon mutant failed to persist in the chronic phase of mouse infection (*Goodsmith et al., 2015*). To distinguish whether this was the result of slowed bacterial replication or enhanced killing by the host, we constructed a *perM* deletion strain (Δ*perM*) (*Xu et al., 2017*) and determined cell replication rates of WT and the Δ*perM* mutant using the 'replication clock' plasmid (*Gill et al., 2009*). Measurements of growth and plasmid loss showed that the two strains replicated at a similar rate in standard in vitro growth conditions (*Figure 1—figure supplement 1*).

During the initial two weeks after aerosol infection with approximately 1000 CFU, WT and Δ*perM* grew logarithmically in mouse lungs (*Figure 1A*). As expected, this was accompanied by a decrease in the frequency of bacteria carrying the 'replication clock' plasmid (*Figure 1B*). The rate of plasmid loss was similar between WT and Δ*perM* (*Figure 1B*), resulting in a calculated doubling time of around 20 hr for both strains (*Figure 1—figure supplement 2A*). During the first two weeks of infection, cell replication rates exceeded death rates in both Mtb strains and led to an increase in bacterial burden (*Figure 1—figure supplement 2B*).

From 2 to 12 weeks post infection, WT Mtb persisted with stable bacterial titers, while Δ*perM* titers decreased (*Figure 1A*). Both strains lost the 'replication clock' plasmid more slowly than during the acute phase of infection, but plasmid loss was significantly slower in Δ*perM* than in WT (*Figure 1B*). The calculated doubling time of Δ*perM* was 280 ± 61 hr, more than twice the doubling time of WT, which was 111 ± 8 hr (*Figure 1—figure supplement 2A*). Consistently, the cell replication rate of Δ*perM* was significantly lower than that of WT while death rates were comparable (*Figure 1—figure supplement 2B*). Next, we quantified the cumulative bacterial burden (CBB), defined as the total number of bacteria in the mouse lung including live, dead and those cleared by host immunity (*Gill et al., 2009*). We found that the CBB of WT increased steadily during weeks 2–12. In marked contrast, the CBB of Δ*perM* increased only slightly during the same time interval (*Figure 1C*). A second mouse infection with approximately 100 CFU administered by aerosol yielded similar results (*Figure 1—figure supplement 3*). Overall, these data demonstrate that the failure of Δ*perM* to persist in the chronic phase of mouse infection can be attributed to a cell replication defect.

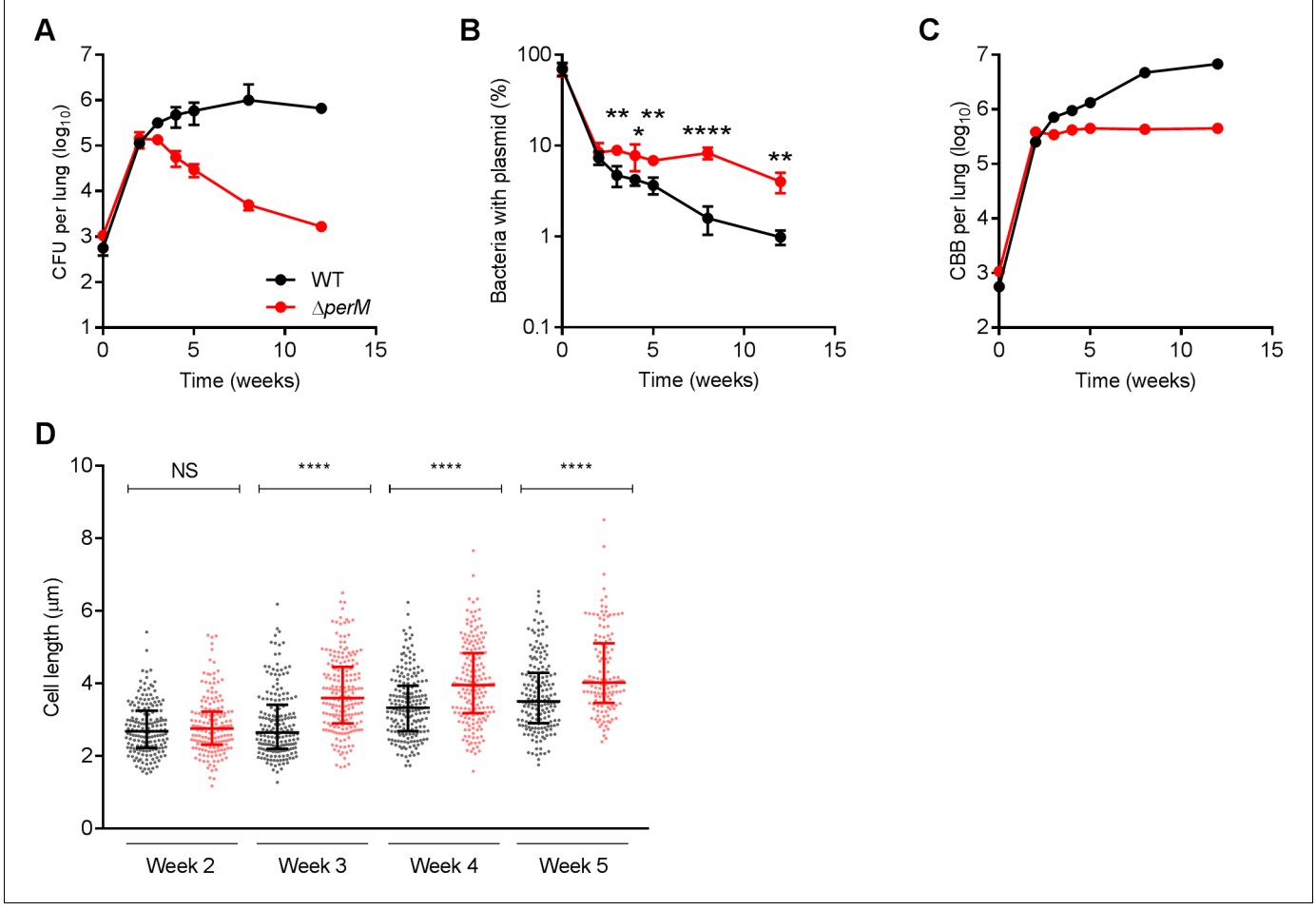

**Figure 1.** Mtb requires PerM for optimal cell division during chronic mouse infection. (A) Bacterial titers in lungs of C57BL/6 mice infected with WT Mtb or the ΔperM mutant containing the replication clock plasmid pBP10 at the indicated time points post-infection. Data are means ± SD of five mice. (B) Fractions of bacteria containing the pBP10 plasmid at the indicated time points. Data are means ± SD of five mice. P-values were calculated using multiple t tests and adjusted for multiple comparisons. *, adj-P <0.05; **, adj-P <0.01; ****, adj-P <0.0001. (C) Calculated cumulative bacterial burden (CBB) in mouse lungs. (D) Scatter plots of cell length of WT Mtb (black) or the ΔperM mutant (red) in acid-fast stained lung sections at the indicated time points. The middle lines represent the medians and the top and bottom lines represent the 25th and 75th percentiles. P-values were calculated using Kruskal-Wallis test and corrected for multiple comparisons. ****, adj-P <0.0001. Data in (A–C) are from one experiment, and data in (D) are representative of two independent experiments.

The online version of this article includes the following source data and figure supplement(s) for figure 1:

**Source data 1.** Summary statistics of *Figure 1D*.
**Figure supplement 1.** Mtb WT and ΔperM replicate similarly in vitro.
**Figure supplement 2.** Mtb ΔperM replicates more slowly than WT in the chronic phase of mouse infection.
**Figure supplement 3.** Replication clock experiment with low Mtb inoculum reveals slower cell replication of ΔperM than WT in the chronic phase of mouse infection.

## PerM facilitates cell division in Mtb during chronic mouse infection

We quantified bacterial cell length in acid fast-stained infected lung sections. At week two, WT and ΔperM had comparable median cell lengths (*Figure 1D*). At week three, ΔperM cells were significantly longer than WT cells and this difference in cell length remained at weeks four and five. These measurements strongly suggest that the observed cell replication defect is associated with impaired cell division during the chronic phase of mouse infection.

## Depletion of PerM leads to defects in septum formation and resolution in *M. smegmatis*

To explore how PerM facilitates cell division, we studied its homolog in *M. smegmatis*, a non-pathogenic mycobacterium; MSMEG_5517 (PerM$_{msm}$) shares 73% identity with PerM from Mtb. Unexpectedly, we failed to isolate a *perM$_{msm}$* deletion mutant, which suggested that *perM$_{msm}$* is essential for growth of *M. smegmatis*. We therefore constructed a PerM dual-control (DUC) mutant (***Figure 2—figure supplement 1A–C***), in which PerM expression is controlled by both transcriptional silencing and proteolytic degradation in response to anhydrotetracycline (atc) (***Kim et al., 2013***; ***Schnappinger et al., 2015***). The number of viable cells remained constant in the PerM-depleted population, indicating that PerM depletion was bacteriostatic over the course of the experiment (***Figure 2—figure supplement 1D***). PerM depletion resulted in an increase of median cell length from 4.46 μm to 11.1 μm in *M. smegmatis* (***Figure 2A,B***). Additionally, peptidoglycan labeling with the fluorescent D-alanine analog HCC-amino-D-alanine (HADA) revealed that the elongated cells formed filaments, with the majority containing either none or only one septum (***Figure 2A,C***) (***Kuru et al., 2012***; ***Kuru et al., 2015***). DNA staining with SYTO 13 revealed separated nucleoids spanning the entire length of each bacterium, suggesting that PerM depletion did not affect DNA segregation (***Figure 2—figure supplement 2***). The phenotypes associated with PerM depletion in *M. smegmatis* demonstrate that PerM is involved in septation.

We then monitored septum formation and resolution in *M. smegmatis* by sequential peptidoglycan labeling with HADA and a second fluorescent D-alanine analog, NBD-amino-D-alanine (NADA) (***Figure 3A***). As expected, incubation of *perM*-DUC with atc led to increased cell length (***Figure 3B, C***); the median cell length increased from 3.79 μm (no atc) to 4.598 μm, 5.191 μm and 6.948 μm after 1.5, 4 and 10 hr of atc treatment. Next, we quantified the formation of new septa (labeled with

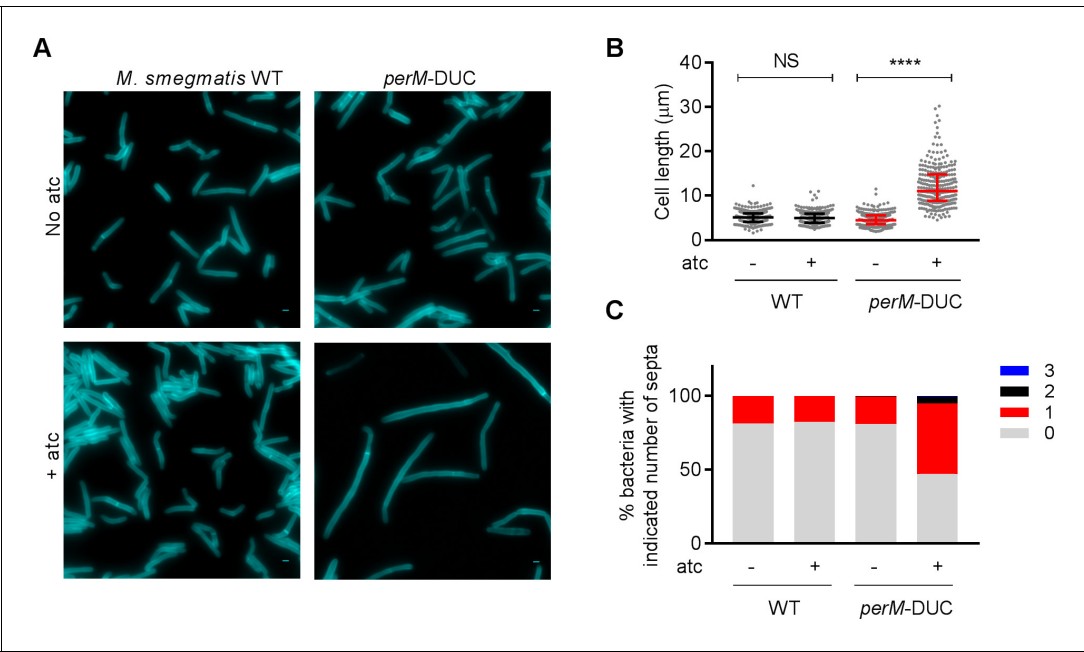

**Figure 2.** Depletion of PerM leads to a septation defect in *M. smegmatis*. (**A**) Representative microscopy images of WT *M. smegmatis* and the *perM*-DUC mutant after 9 hr of incubation in 7H9 medium supplemented with 1 mM HADA and in the absence or presence of 200 ng/ml atc. Scale bar, 1 μm. (**B**) Scatter plots of bacterial cell length of *M. smegmatis* strains from (**A**). The middle lines represent the medians and the top and bottom lines represent the 25[th] and 75[th] percentiles. P-values were computed using Kruskal-Wallis test and adjusted for multiple comparisons. ****, adj-P <0.0001. (**C**) Quantification of bacterial cells that contain 0, 1, 2 or 3 septa of *M. smegmatis* strains from (**A**). Data in (**A–C**) are representative of two independent experiments.

The online version of this article includes the following source data and figure supplement(s) for figure 2:

**Source data 1.** Summary statistics of *Figure 2B*.
**Figure supplement 1.** Construction and characterization of a PerM$_{msm}$ depletion mutant (*perM*-DUC).
**Figure supplement 2.** Nucleoid morphology of WT *M. smegmatis* and *perM*-DUC in the absence and presence of atc.

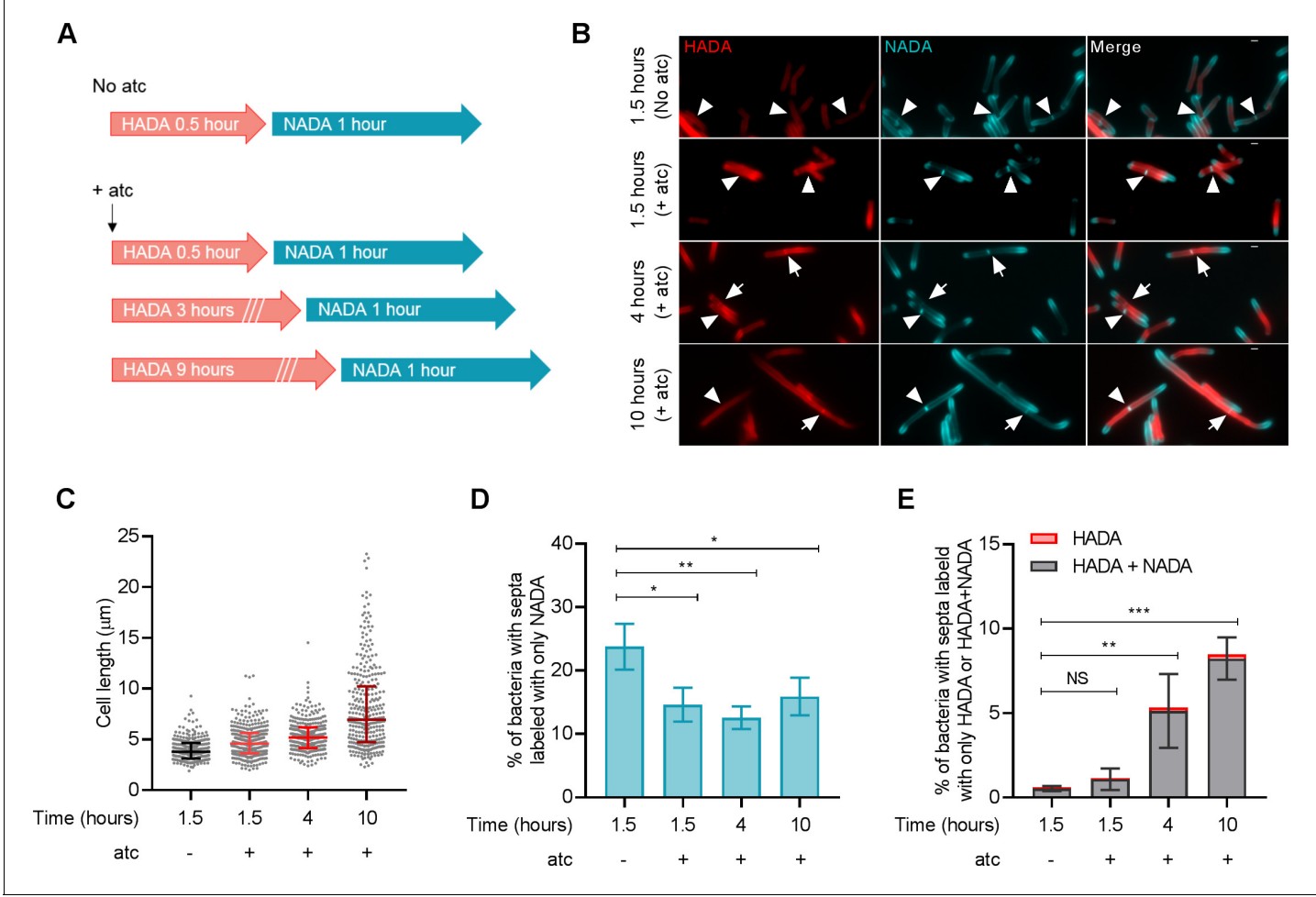

**Figure 3.** PerM depletion leads to defects in septum formation and resolution in *M. smegmatis*. (A) Schematic of peptidoglycan labeling of *M. smegmatis perM*-DUC. Bacterial cells were incubated in growth medium containing 1 mM HADA and 200 ng/ml atc for 0.5, 3, or 9 hr. Cells were then washed to remove excess HADA and incubated with 1 mM NADA and 200 ng/ml atc for 1 hr. *PerM*-DUC cells incubated in growth medium containing D-alanine probes but without atc were used as controls. (B) Representative microscopy images of *perM*-DUC after the sequential peptidoglycan labeling. Arrowheads point to septa labeled with NADA only; and arrows point to septa co-labeled with HADA and NADA. Scale bar, 1 μm. (C) Scatter plots of *perM*-DUC cell lengths from (B). Lines represent median and the 25th and 75th percentiles. Data are representative of three biological replicates. (D) Proportion of bacteria that contain septa labeled exclusively with NADA. Lines indicate means ± SD of three independent experiments. (E) Proportion of bacteria that contain septa labeled exclusively with HADA or co-labeled with NADA. Lines indicate means ± SD of septa labeled with HADA+NADA from three independent experiments. Statistics are reported on the HADA+NADA datasets. For data in (D and E), P-values were computed using ANOVA and adjusted for multiple comparisons. *, adj-P <0.05; **, adj-P <0.01; ***, adj-P <0.001.

The online version of this article includes the following source data for figure 3:

**Source data 1.** Summary statistics of *Figure 3C*.
**Source data 2.** Summary of *Figure 3D and 3E*.

NADA alone) and the resolution of old septa (labeled with HADA alone or both D-alanine probes) after PerM depletion. The percentage of septa labeled with NADA alone declined from 23.77% (no atc) to 14.62%, 12.56% and 15.92% after 1.5, 4 and 10 hr incubation with atc (*Figure 3D*), suggesting that PerM-depleted cells had an impaired ability to form new septa. In contrast, the proportion of HADA-containing septa increased with time, from 0.61% (no atc) to 1.14%, 5.34% and 8.48% after 1.5, 4 and 10 hr atc treatment (*Figure 3E*). Collectively, these results indicate that PerM depletion affects both septum formation and cell separation in *M. smegmatis*, causing the bacteria to form filaments.

## Ectopic FtsB expression rescues the phenotypes caused by PerM depletion in *M. smegmatis* and PerM deletion in Mtb

Septation requires the coordinated activities of cell wall biosynthetic and lytic enzymes (*Kieser and Rubin, 2014*). To identify proteins that might rescue the septation defects caused by PerM depletion in *M. smegmatis*, we performed a forward genetic screen. We transformed *perM*-DUC with an Mtb genomic library and selected for colonies able to grow on agar containing atc, a non-growth-permissive condition for the *perM*-DUC mutant. The plasmids that rescued growth of *perM*-DUC contained either *rv0955* (*perM*) or *rv1024*.

*Rv1024* encodes the Mtb homolog of FtsB (FtsB$_{mtb}$), an integral membrane protein of the bacterial divisome that is conserved across different bacterial species (*Buddelmeijer et al., 2002*; *Levin and Losick, 1994*; *Wu et al., 2018*). In model bacteria such as *E. coli*, FtsB is recruited to the septum as part of a complex that also contains FtsQ and FtsL, two other bitopic proteins essential for cell division (*Buddelmeijer and Beckwith, 2004*; *Glas et al., 2015*). The FtsQLB complex is thought to drive septation by stimulating peptidoglycan synthesis by the peptidoglycan synthases FtsI and FtsW along with their associated proteins (*Kureisaite-Ciziene et al., 2018*; *Liu et al., 2015*; *Taguchi et al., 2019*; *Tsang and Bernhardt, 2015*). *M. smegmatis* encodes homologs of all three proteins. Like in *E. coli*, depletion of FtsQ, FtsL or FtsB in *M. smegmatis* resulted in morphological changes indicative of impaired septation (*Buddelmeijer et al., 2002*; *Carson et al., 1991*; *Guzman et al., 1992*; *Wu et al., 2018*).

FtsB$_{mtb}$ is 72% identical to FtsB$_{msm}$, and ectopic expression of either protein from a multicopy plasmid rescued the growth defect caused by PerM depletion in *M. smegmatis* (*Figure 4A*). Furthermore, ectopically-expressed FtsB$_{mtb}$ or FtsB$_{msm}$ reversed the cell elongation phenotype as did complementation with *perM*$_{mtb}$ or *perM*$_{msm}$ (*Figure 4B*). Therefore, PerM likely acts on FtsB to facilitate septum formation in *M. smegmatis*. Moreover, FtsB$_{mtb}$ and FtsB$_{msm}$ appear to be functionally conserved.

As we reported previously (*Goodsmith et al., 2015*), PerM is required for cell replication and survival of Mtb in media with reduced magnesium (Mg$^{2+}$) and Mtb lacking PerM elongated in response to Mg$^{2+}$ limitation (*Figure 4C,D*). Both the growth and elongation phenotypes were, to a large extent, rescued by ectopic expression of FtsB, indicating that PerM likely functions via FtsB in Mtb as well.

## PerM interacts with FtsB

To assess the molecular mechanism by which PerM contributes to cell division and to determine if FtsB and PerM interact physically, we immunoprecipitated FtsB$_{mtb}$ containing an N-terminal Flag-tag from Mtb protein lysates and identified its interactors using mass spectrometry (*Table 1*). PerM$_{mtb}$ co-immunoprecipitated with FtsB$_{mtb}$, suggesting that the two proteins are part of the same complex. This interaction was verified in *M. smegmatis* strains that co-expressed PerM$_{mtb}$ and FtsB$_{mtb}$ with different protein tags (*Figure 5—figure supplement 1*). The mass spectrometry data also revealed that FtsB interacts with FtsQ in Mtb (*Table 1*). Similar to its homologs in *E. coli* and *B. subtilis*, FtsQ$_{mtb}$ has been shown to be a septal-localizing integral membrane protein that modulates cell division (*Buddelmeijer and Beckwith, 2004*; *Jain et al., 2018*; *Katis et al., 2000*). FtsB, FtsL and FtsQ form a complex in *E. coli* (*Buddelmeijer and Beckwith, 2004*; *Kureisaite-Ciziene et al., 2018*), and here we provide evidence that the interaction between FtsB and FtsQ is conserved in mycobacteria.

## FtsB localizes to the mid-cell independently of PerM

We next investigated the spatiotemporal dynamics of FtsB and PerM during the cell cycle by co-expressing FtsB$_{mtb}$ and PerM$_{mtb}$ fused to different fluorescent proteins in *M. smegmatis*. Time-lapse microscopy showed that FtsB localized to the mid-cell ahead of PerM (*Figure 5A*). PerM localized to the same site approximately 15 min later with both proteins present at the mid-cell prior to septum constriction. We quantified the fluorescence intensities at the mid-cell and observed that the FtsB signals started to increase at ~30% and peaked at ~50% of the cell cycle, whereas the PerM signals increased at ~40% and stabilized by ~50% of the cell cycle (*Figure 5B*), confirming that FtsB precedes PerM at the division site. This sequential localization was not an artifact caused by the fluorescent tags, as exchanging the tags resulted in the same order, with FtsB detectable at the mid-cell

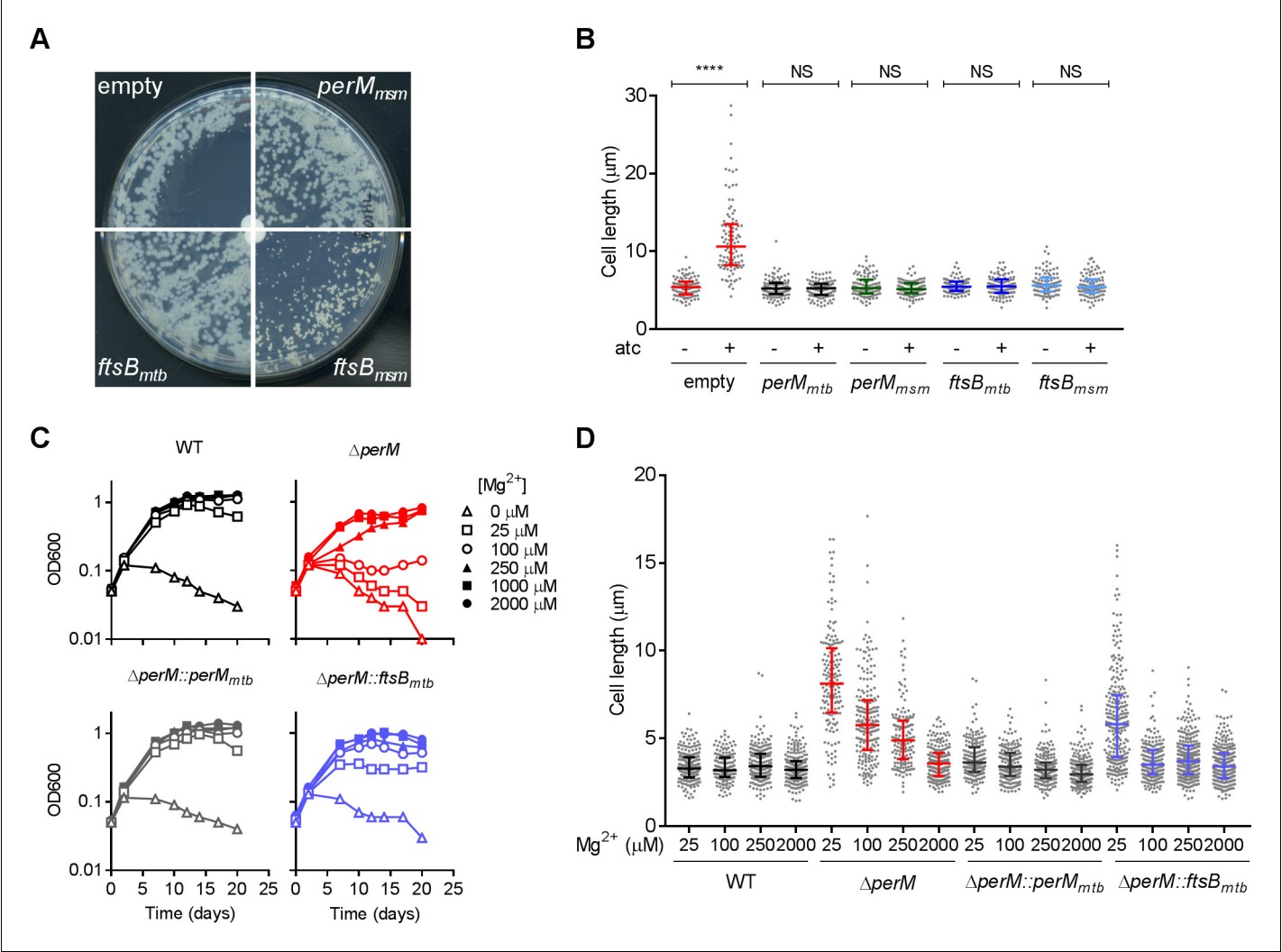

**Figure 4.** Ectopic *ftsB* expression rescues the phenotypes caused by PerM depletion in *M. smegmatis* and *perM* deletion in Mtb. (**A**) *M. smegmatis perM*-DUC transformed with a control plasmid or plasmids that express *perM_{msm}*, *ftsB_{mtb}* or *ftsB_{msm}* under the constitutive hsp60 promoter were cultured on 7H10 agar. The paper disks in the center of the plates contained 50 ng atc, which diffused and created an atc gradient from the center to the periphery. Images of plates were taken on day 4 (*perM*-DUC-control, *perM*-DUC-*perM_{msm}* and *perM*-DUC-*ftsB_{mtb}*) or day 7 (*perM*-DUC-*ftsB_{msm}*). (**B**) Scatter plots of the lengths of *perM*-DUC-control, *perM*-DUC-*perM_{mtb}*, *perM*-DUC-*perM_{msm}*, *perM*-DUC-*ftsB_{mtb}* and *perM*-DUC-*ftsB_{msm}* incubated in 7H9 medium without and with atc for 24 hr. The lines represent the $25^{th}$, $50^{th}$ and $75^{th}$ percentiles. P-values were computed using Kruskal-Wallis test and adjusted for multiple comparisons. ****, adj-P <0.0001. (**C**) Growth curves of Mtb incubated in chelated Sauton's medium supplemented with $Mg^{2+}$ at the indicated concentrations. (**D**) Scatter plots of Mtb cell lengths after 5 days of incubation in Sauton's medium containing $Mg^{2+}$ at the indicated concentrations. The lines represent the $25^{th}$, $50^{th}$ and $75^{th}$ percentiles. P-values were computed using Kruskal-Wallis test and corrected for multiple comparisons. The adjusted P-values are <0.0001 for the following comparisons: WT with Δ*perM* at 25 μM, 100 μM, 250 μM $Mg^{2+}$, WT with Δ*perM*::*ftsB_{mtb}* at 25 μM $Mg^{2+}$; Δ*perM* with Δ*perM*::*ftsB_{mtb}* at 25 μM $Mg^{2+}$. Adj-P = 0.0762, 0.1095 and 0.7284 comparing WT to Δ*perM*::*ftsB_{mtb}* at 100 μM, 250 μM and 2000 μM $Mg^{2+}$, respectively. Data in (**A–C**) are representative of two independent experiments, and data in (**D**) are from one experiment.

The online version of this article includes the following source data for figure 4:

**Source data 1.** Summary statistics of *Figure 4B*.
**Source data 2.** Summary statistics of *Figure 4D*.

before PerM (*Figure 5—figure supplement 2A*). In addition, in *M. smegmatis* lacking PerM but ectopically expressing a second copy of FtsB, FtsB also localized to the septum (*Figure 5—figure supplement 2B*). This suggests that the localization of FtsB to mid-cell does not depend on the presence of PerM.

**Table 1.** Mass spectrometry identification of protein interaction partners of FtsB in Mtb.

| Rv # | Gene | Annotation | Sum total spectrum count |
|---|---|---|---|
| Rv1687c | | ABC transporter ATP-binding protein | 20 |
| Rv2901c | | Hypothetical protein | 16 |
| Rv1280c | oppA | Probable periplasmic oligopeptide-binding lipoprotein OppA | 16 |
| Rv2151c | ftsQ | Cell division protein FtsQ | 15 |
| Rv1697 | | Hypothetical protein | 14 |
| Rv3330 | dacB1 | Probable penicillin-binding protein DacB1 | 14 |
| Rv1698 | mctB | Outer membrane protein MctB | 14 |
| Rv1266c | pknH | Serine/threonine protein kinase | 12 |
| Rv2748c | ftsK | Cell division protein FtsK | 11 |
| Rv0955 | perM | | 11 |
| Rv0050 | ponA1 | Peptidoglycan glycosyltransferase | 10 |
| Rv3493c | | MCE-associated alanine and valine rich protein | 10 |
| Rv0497 | | Probable conserved transmembrane protein | 10 |
| Rv3212 | | Conserved alanine valine rich protein | 9 |
| Rv0046c | ino1 | Inositol-1-phosphate synthase | 8 |

Flag-FtsB$_{mtb}$ was immunoprecipitated from whole-cell lysates of Mtb. Whole-cell lysates of an Mtb strain constitutively expressing only the Flag-tag served as control. The protein interaction partners were identified by mass spectrometry and data are from two independent biological replicates.

The online version of this article includes the following source data for Table 1:

**Source data 1.** Mass spectrometry data of FtsB pulldown in Mtb.

## PerM stabilizes FtsB

FtsB protein levels were markedly reduced in MtbΔ*perM* relative to WT or the complemented mutant, despite similar *ftsB* mRNA levels (*Figure 5C,D*, *Figure 5—figure supplement 3A*). We therefore tested whether PerM stabilizes FtsB by monitoring FtsB levels after blocking protein synthesis with chloramphenicol. In both WT Mtb and the complemented mutant, FtsB amounts were stable over 96 hr (*Figure 5E*, *Figure 5—figure supplement 3B,C*). In contrast, FtsB abundance decreased over time in Δ*perM*. Because FtsB levels were low in Δ*perM* and thus difficult to assess, we also evaluated FtsB stability in Δ*perM* containing an additional copy of *ftsB* (Δ*perM::ftsB*$_{mtb}$) resulting in increased FtsB levels. In the absence of PerM, the abundance of FtsB still decreased over time. These results demonstrate that PerM stabilizes FtsB.

## PerM is required for normal growth and cell division of Mtb in host-relevant stress conditions

MtbΔ*perM* was specifically attenuated during the chronic phase of mouse infection following the onset of adaptive immunity when T-cell derived IFN-γ activates macrophages to control Mtb (*Flynn et al., 1993*). In IFN-γ-activated macrophages, Mtb faces acidic pH, reactive oxygen and nitrogen intermediates and limited amounts of iron and other nutrients (*Ehrt and Schnappinger, 2009*; *Stallings and Glickman, 2010*). We reported previously that an Mtb *perM* transposon mutant survived these stresses similarly to WT in vitro (*Goodsmith et al., 2015*), consistent with our finding that the death rates of Δ*perM* and WT were similar during the chronic phase of mouse infection (*Figure 1—figure supplement 2B*). Because the growth rate of Δ*perM* was significantly reduced compared to that of WT during chronic infection, we tested whether PerM is required to sustain bacterial replication under host-relevant stress conditions, such as acidic conditions and iron-restricted medium. In contrast to its near normal growth at pH 7, Δ*perM* exhibited a growth defect and elongated cell morphology at pH 5.5 (*Figure 6A–C*). Ectopically-expressed FtsB reversed the growth defect and cell elongation phenotype, suggesting that both were caused by insufficient amounts of FtsB. Notably, FtsB levels were similar at pH 7 and pH 5.5, suggesting an increased demand for FtsB during growth at low pH (*Figure 6D*). Growth in iron-depleted medium resulted in

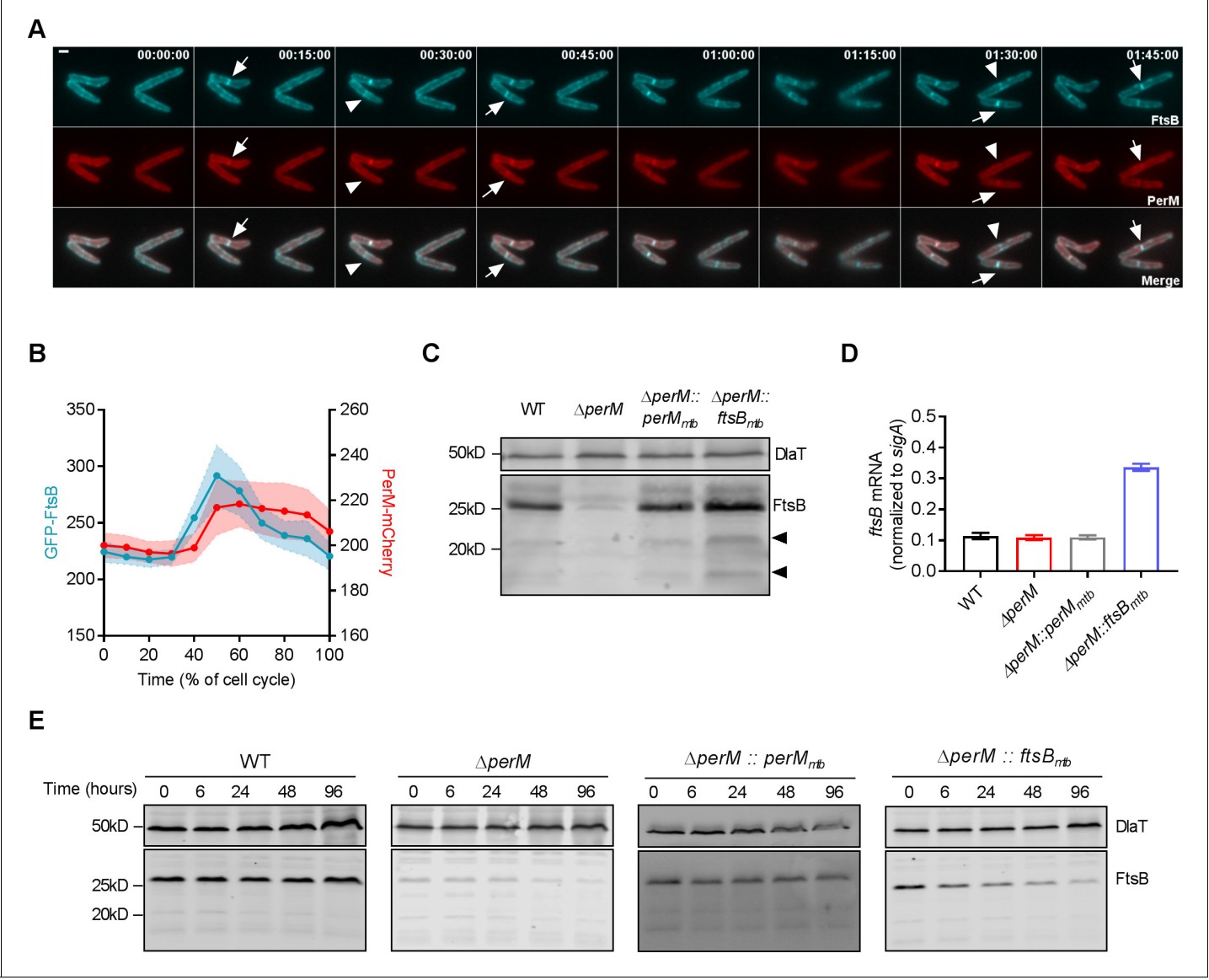

**Figure 5.** PerM co-localizes with FtsB and stabilizes FtsB. (A) Representative image series from time lapse movies of replicating *M. smegmatis* constitutively expressing both GFP-FtsB$_{mtb}$ and PerM$_{mtb}$-mCherry over the duration of 1 hr and 45 mins. Arrows point to the presence of both FtsB and PerM at the mid-cell, and arrowheads point to the presence of FtsB and lack of PerM at the mid-cells. Scale bar, 1 μm. Numbers in the upper right corner indicate time. (B) Measurements of the maximum fluorescence intensities of GFP-FtsB and PerM-mCherry as a function of the cell cycle from time lapse movies shown in (A). The lines and shaded areas indicate means and the 95% confidence intervals of 20 bacteria. (C) FtsB protein in whole-cell lysates of log-phase Mtb measured by western blotting with anti-FtsB antibody. Arrowheads point to cleavage products of FtsB. Dihydrolipoamide acyltransferase (DlaT) was used as loading control. (D) *FtsB* mRNA in log-phase Mtb culture measured by quantitative real time PCR. mRNA levels were normalized to expression of the housekeeping gene *sigA*. Data are means ± SD of triplicates. (E) Detection of FtsB in whole-cell lysates collected from Mtb strains by western blotting with anti-FtsB antibody. 20 μg/ml chloramphenicol were added to each culture at time 0 to inhibit protein synthesis and samples were collected at the indicated time points. Western blotting with anti-DlaT was used as loading control. Data in (C–E) are representative of two independent experiments. Biological replicates of *Figure 5C and E* are shown in *Figure 5—figure supplement 3*.
The online version of this article includes the following figure supplement(s) for figure 5:

**Figure supplement 1.** FtsB$_{mtb}$ and PerM$_{mtb}$ interact in vivo.
**Figure supplement 2.** Localization of FtsB to the mid-cell does not depend on the presence of PerM.
**Figure supplement 3.** PerM stabilizes FtsB.

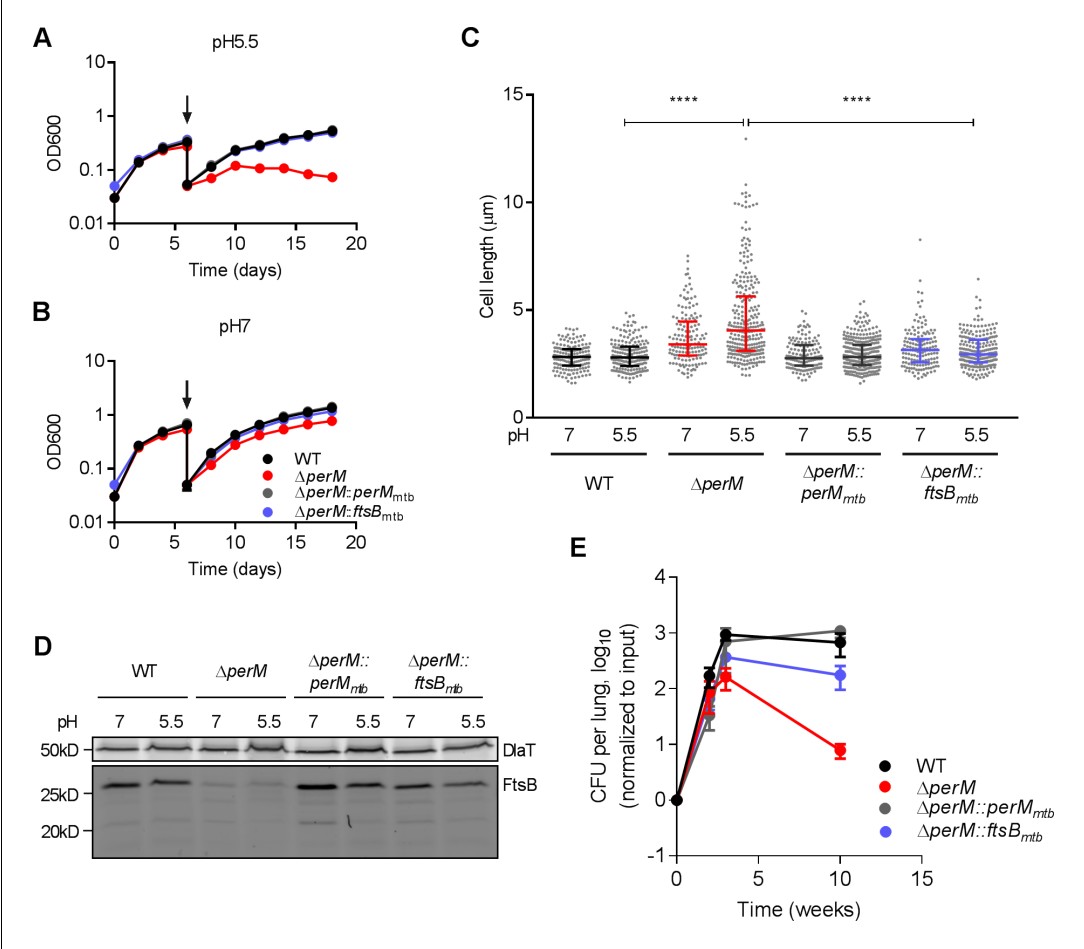

**Figure 6.** PerM is required for normal growth and cell division of Mtb in host-relevant stress conditions. (**A,B**) Growth curves of Mtb strains sequentially cultured in 7H9 medium adjusted to either pH 5.5 (**A**) or pH 7 (**B**). At day 6 (arrows), the cultures were diluted in the same medium and samples collected for length quantification. Data are means ± SD of three replicates. (**C**) Scatter plots of the cell length of Mtb 6 days post incubation at either pH 5.5 or pH 7. The middle lines represent the medians and the bottom and top lines represent the 25th and 75th percentiles. P-values were calculated using Kruskal-Wallis test and corrected for multiple comparisons. ****, adj-P <0.0001. (**D**) FtsB levels in whole-cell protein lysates assayed by western blotting with anti-FtsB antibody. Whole-cell protein lysates were collected from Mtb cultured in 7H9 medium adjusted to either pH 7 or pH 5.5 for 3 or 6 days. DlaT was used as loading control. (**E**) Bacterial titers in lungs of C57BL/6 mice infected with indicated Mtb strains. Data are means ± SD of four mice and normalized to input. Data in (**A, B and D**) are representative of two independent experiments, and data in (**C and E**) are from one experiment. The online version of this article includes the following source data and figure supplement(s) for figure 6:

**Source data 1.** Summary statistics of *Figure 6C*.
**Figure supplement 1.** PerM is required for normal growth and cell division of Mtb in low-iron medium.
**Figure supplement 1—source data 1.** Summary statistics of *Figure 6—figure supplement 1B*.

phenotypic changes similar to those observed at low pH, and these could also be reversed by ectopic FtsB expression (*Figure 6—figure supplement 1*). Therefore, in host-relevant stress conditions, PerM is required to maintain FtsB at levels that are sufficient for growth and cell division.

To extend these findings to the in vivo environment, we infected mice with Δ*perM::ftsB*mtb and found that ectopic FtsB expression rescued the persistence defect of Δ*perM* (*Figure 6E*). Thus, PerM-mediated FtsB stabilization is required for growth and cell division in vivo, specifically in the chronic phase of infection.

Taken together, these results indicate that PerM, a member of the mycobacterial divisome, stabilizes FtsB and maintains it at levels sufficient for cell division in the host-imposed stress conditions that characterize the chronic phase of mouse infection. Our results additionally highlight that Mtb

cell division during the chronic phase of mouse infection depends on specific divisome components that differ from those required during the acute phase of infection.

## Discussion

Subclinical persistent infections caused by actively-growing bacteria have been described for multiple bacterial pathogens (*Certain et al., 2017*; *Fisher et al., 2017*). In individuals with latent TB infection, bacterial populations are heterogeneous, exhibit differences in metabolic state and susceptibility to antibiotics, and are likely to include actively-replicating subpopulations (*Aldridge et al., 2012*; *Cadena et al., 2017*; *Esmail et al., 2016*; *Malherbe et al., 2016*). In mice Mtb continues to replicate during chronic infection (*Gill et al., 2009*), but little is known about the required divisome components. We found that PerM is part of the mycobacterial divisome and facilitates cell division by stabilizing FtsB during chronic mouse infection. FtsB is a member of the divisome in many bacterial species. In *E. coli*, FtsB is associated with two other bitopic membrane proteins, FtsQ and FtsL, and promotes septation and cell constriction (*Buddelmeijer and Beckwith, 2004*; *Glas et al., 2015*), (*Goehring and Beckwith, 2005*; *Liu et al., 2015*; *Tsang and Bernhardt, 2015*). Mycobacterial FtsQ, FtsL and FtsB are structurally conserved and functionally similar to their *E. coli* homologs (*Jain et al., 2018*; *Wu et al., 2018*); however, regulation of FtsB differs between the two species. In *E. coli*, the stability of FtsB depends on FtsL and vice versa (*Gonzalez and Beckwith, 2009*), while we found that PerM is required to stabilize FtsB in Mtb; FtsL alone does not protect FtsB from degradation in the absence of PerM. PerM is highly conserved among mycobacteria and other actinobacteria but has no homologs outside of this group, implying its function may be specific to actinobacteria. Within mycobacteria, we have shown that PerM encoded by Mtb and *M. smegmatis* are conserved in function (*Figure 4B*), but the two proteins differ in essentiality for growth under standard experimental conditions. This difference may be due to a varied demand for FtsB during cell division in the sister species. Immunoprecipitation experiments revealed that PerM physically interacts with FtsB (*Table 1*, *Figure 5—figure supplement 1*), although it is not clear whether this represents direct binding, or an interaction mediated by other proteins. Nonetheless, PerM may protect FtsB from recognition or access by an unidentified protease. A similar scenario has been observed in *B. subtilis*, the FtsB homolog DivIC interacts with FtsL, preventing proteolysis of FtsL by the intramembrane protease RasP (*Wadenpohl and Bramkamp, 2010*). Alternatively, PerM may stabilize FtsB and possibly the FtsQLB complex (*Wu et al., 2018*) by facilitating protein-protein interactions within the divisome.

What could be the benefit of expressing an inherently unstable protein such as FtsB along with a stabilizing factor such as PerM, as opposed to simply expressing a more stable form of the former? Examples of other cell division proteins suggest that their inherent instability is important for temporal control of cell division. In *B. subtilis*, FtsL degradation by RasP has been proposed to occur after the cell cycle has been completed and the divisome disassembled, thus preventing reassembly of the divisome and subsequent cell division at the wrong time (*Wadenpohl and Bramkamp, 2010*). In *E. coli*, the concentration of FtsZ has been shown to oscillate during the cell cycle in slow growth conditions and FtsZ is degraded by the ClpXP protease at the end of the cell cycle (*Männik et al., 2018*). Further, the amplitude of FtsZ oscillation increased in conditions where growth is slowed, indicating that growth conditions may impact the regulation of cell division. In Mtb, the expression of an unstable FtsB along with its stabilizer PerM may allow for fine-tuning of FtsB levels during the cell cycle, which could be important in adaptation to the host environment. Consistent with this hypothesis, Δ*perM* showed growth and cell division defects in stress conditions mimicking the intracellular environment.

The abundance of FtsB was not significantly changed upon exposure of the bacteria to stress despite a marked reduction in cell replication rate. This observation indicates that either FtsB activity is altered under stress conditions or that the divisome formed in Mtb under stress conditions differs from the standard log-phase divisome, displaying an increased demand for FtsB. These hypotheses are supported by studies in mycobacteria that showed alterations in activities or protein-protein interactions of divisome components in response to stress. For example, MarP, a periplasmic protease, interacts more strongly with the peptidoglycan hydrolase RipA in acidic pH, resulting in RipA activation (*Botella et al., 2017a*). In addition, FtsZ requires interaction with FipA for mid-cell localization and Z-ring formation in *M. smegmatis* during oxidative stress, but not in normal growth

conditions (*Sureka et al., 2010*). Z-ring formation has also been shown to be compromised in intracellular Mtb, likely due to altered FtsZ stability or activity in stresses imposed by the intracellular environment (*Chauhan et al., 2006*).

The environmental stimuli regulating cell division proteins are often transduced by protein kinases (*Molle and Kremer, 2010*). In addition to PerM and FtsQ, our co-immunoprecipitation experiment revealed interaction of FtsB with PknH, a Ser/Thr protein kinase involved in regulation of cell division (*Sharma et al., 2006*; *Zheng et al., 2007*) (*Table 1*). Other FtsB interactors include cell division-related proteins such as DacB1, FtsK and PonA1; an ATP-binding protein; membrane proteins; an inositol 1-phosphate synthase; transporters; and proteins of unknown function. Future research will aim to delineate the pathways connecting environmental changes to FtsB regulation in Mtb.

In summary, this work provides new insights into the regulation of cell division in Mtb and its importance for persistence in the host. The cell division and in vivo persistence defects caused by inactivating PerM, a protein dispensable for bacterial replication in standard in vitro growth conditions, reveal that the requirements for cell division differ in vitro between standard growth conditions and host-relevant stress conditions, as well as in vivo between acute and chronic phases of mouse infection. Whether sustained cell replication also facilitates persistence of Mtb in humans is as yet unknown, but evidence for ongoing bacterial transcription and bacterial heterogeneity during latent TB infection support this hypothesis (*Esmail et al., 2016*; *Malherbe et al., 2016*). Given the importance of cell division for persistence of Mtb in chronically-infected mice and its potential importance for persistence in latently-infected humans, future work will identify and characterize cell division factors that are specifically required for persistence in vivo. This will not only provide insights into host-pathogen interactions, but also lay the groundwork for potential strategies to combat persistent Mtb infections.

# Materials and methods

## Key resources table

| Reagent type (species) or resource | Designation | Source or reference | Identifiers | Additional information |
|---|---|---|---|---|
| Strain, strain background (*Mycobacterium tuberculosis*) | WT | This work | | H37Rv |
| Strain, strain background (*Mycobacterium tuberculosis*) | $\Delta perM$ | DOI: 10.1128/AAC.01334-17 | | H37Rv |
| Strain, strain background (*Mycobacterium tuberculosis*) | $\Delta perM::perM_{mtb}$ | This work | | H37Rv $\Delta perM::$ hsp60-$perM_{mtb}$ |
| Strain, strain background (*Mycobacterium tuberculosis*) | $\Delta perM::ftsB_{mtb}$ | This work | | H37Rv $\Delta perM::$hsp60-$ftsB_{mtb}$ |
| Strain, strain background (*Mycobacterium tuberculosis*) | WT-pBP10 | This work Plasmid information in DOI: 10.1038/nm.1915 | | H37Rv pBP10 plasmid obtained from Dr. David Sherman (University of Washington) |
| Strain, strain background (*Mycobacterium tuberculosis*) | $\Delta perM$-pBP10 | This work Plasmid information in DOI: 10.1038/nm.1915 | | H37Rv pBP10 plasmid obtained from Dr. David Sherman (University of Washington) |

*Continued on next page*

Continued

| Reagent type (species) or resource | Designation | Source or reference | Identifiers | Additional information |
|---|---|---|---|---|
| Strain, strain background (*Mycobacterium tuberculosis*) | WT::hsp60-*flag* | This work | | H37Rv |
| Strain, strain background (*Mycobacterium tuberculosis*) | WT::hsp60-*flag*-*ftsB*$_{mtb}$ | This work | | H37Rv |
| Strain, strain background (*Mycobacterium smegmatis*) | WT | This work | | mc$^2$155 |
| Strain, strain background (*Mycobacterium smegmatis*) | *perM*-DUC | This work | | mc$^2$155 Δ*perM*::T38S38-P766-9T-*perM*$_{mtb}$-*gfp*-das+4::TSC10M1-SD2-SSPB |
| Strain, strain background (*Mycobacterium smegmatis*) | *perM*-DUC-control | This work | | mc$^2$155 *perM*-DUC + hsp60-empty |
| Strain, strain background (*Mycobacterium smegmatis*) | *perM*-DUC- *perM*$_{mtb}$ | This work | | mc$^2$155 *perM*-DUC + hsp60-*perM*$_{mtb}$-*gfp* |
| Strain, strain background (*Mycobacterium smegmatis*) | *perM*-DUC- *perM*$_{msm}$ | This work | | mc$^2$155 *perM*-DUC + hsp60-*perM*$_{msm}$-*gfp* |
| Strain, strain background (*Mycobacterium smegmatis*) | *perM*-DUC-*ftsB*$_{mtb}$ | This work | | mc$^2$155 *perM*-DUC + hsp60-*ftsB*$_{mtb}$ |
| Strain, strain background (*Mycobacterium smegmatis*) | *perM*-DUC-*ftsB*$_{msm}$ | This work | | mc$^2$155 *perM*-DUC + hsp60-*ftsB*$_{msm}$ |
| Strain, strain background (*Mycobacterium smegmatis*) | Δ*perM*::P38-*perM*$_{mtb}$-*mcherry*::hsp60-*gfp*-*ftsB*$_{mtb}$ | This work | | mc$^2$155 |
| Strain, strain background (*Mycobacterium smegmatis*) | Δ*perM*::P38-*perM*$_{mtb}$-*mcherry*::hsp60-*gfp* | This work | | mc$^2$155 |
| Strain, strain background (*Mycobacterium smegmatis*) | WT::P38-*gfp* + hsp60-*snap*-*ftsB*$_{mtb}$ | This work | | mc$^2$155 |
| Strain, strain background (*Mycobacterium smegmatis*) | WT::P38-*perM*$_{mtb}$-*gfp* + hsp60-*snap*-*ftsB*$_{mtb}$ | This work | | mc$^2$155 |
| Strain, strain background (*Mycobacterium smegmatis*) | Δ*perM*::P38-*perM*$_{mtb}$-*gfp*::hsp60-*mcherry*-*ftsB*$_{mtb}$ | This work | | mc$^2$155 |

*Continued*

| Reagent type (species) or resource | Designation | Source or reference | Identifiers | Additional information |
|---|---|---|---|---|
| Strain, strain background (*Mycobacterium smegmatis*) | Δ*perM*::P38 -*gfp-ftsB*$_{mtb}$ | This work | | mc$^2$155 |
| Strain, strain background (*Mycobacterium smegmatis*) | Δ*perM*::hsp60 -*gfp-ftsB*$_{mtb}$ | This work | | mc$^2$155 |
| Antibody | Anti-Flag M2 affinity gel | Sigma | A2220 | |
| Antibody | Anti-GFP mAb-agarose beads | MBL | D153-8 | |
| Antibody | Anti-GFP mAb-magnetic beads | MBL | D153-11 | |
| Antibody | Anti-FtsB (Rabbit polyclonal) | GenScript | U1550BG270 | (1:1000) |
| Antibody | Anti-DlaT (Rabbit polyclonal) | DOI: 10.1126/science.1067798 | | Antibody from Drs. Carl Nathan and Ruslana Bryk (Weill Cornell Medicine) (1:10000) |
| Antibody | Anti-GFP (Mouse monoclonal) | Clontech | 632380 | (1:8000) |
| Antibody | Anti-mCherry (Mouse monoclonal) | Clontech | 632543 | (1:1000) |
| Antibody | Anti-SNAP (Rabbit polyclonal) | New England BioLabs | P9310S | (1:1000) |
| Antibody | donkey anti-rabbit 680 | LI-COR Biosciences | 926–68023 | (1:10000) |
| Antibody | donkey anti-rabbit 800 | LI-COR Biosciences | 926–32213 | (1:10000) |
| Antibody | goat anti-mouse 800 | LI-COR Biosciences | 926–32210 | (1:5000) |
| Peptide, recombinant protein | FLAG peptide | Sigma | F3290 | (100 ng/μl) |
| Software, algorithm | Prism 7 | Graphpad | | |
| Software, algorithm | ImageJ | DOI: 10.1038/nmeth.2089 | | |
| Software, algorithm | Icy | DOI: 10.1038/nmeth.2075 | | |
| Software, algorithm | Scaffold 4 | | | |
| Other | SYTO 13 Green Fluorescent Nucleic Acid Stain | Thermo Fisher Scientific | S7575 | (5 μM) |
| Other | HADA | DOI: 10.1002/anie.201206749; DOI: 10.1038/nprot.2014.197 | | (1 mM) Synthesized by the Chemical Synthesis Core Facility at MSKCC |
| Other | NADA | DOI: 10.1002/anie.201206749; DOI: 10.1038/nprot.2014.197 | | (1 mM) Synthesized by the Chemical Synthesis Core Facility at MSKCC |

## Bacterial culture conditions

*M. smegmatis* mc²155 and derivative strains were cultured in Middlebrook 7H9 medium (BD Difco) containing 0.2% glycerol and 0.05% Tween 80 or Middlebrook 7H10 agar (BD Difco) containing 0.5% glycerol. *M. tuberculosis* H37Rv and derivative strains were cultured in Middlebrook 7H9 medium containing 0.2% glycerol, 0.2% dextrose, 0.5% BSA (Roche), 0.05% Tween 80 or tyloxapol, and 0.085% NaCl or Middlebrook 7H10 agar containing 10% OADC supplement (BD) and 0.5% glycerol. Selection antibiotics were used at the following concentrations: hygromycin (50 µg/ml), zeocin (25 µg/ml), kanamycin (25 µg/ml) or streptomycin (20 µg/ml).

## Mutant construction

The MtbΔ*perM* mutant was constructed by allelic exchange using a recombineering approach as previously described (*Gee et al., 2012*). The Δ*perM* strain was complemented by introducing a copy of *perM*, expressed under the control of the hsp60 promoter, into the attL5 site of the Mtb genome. The Δ*perM::ftsB* mutant was constructed by integrating a copy of *ftsB*, expressed under the control of the hsp60 promoter, into the attL5 site of the Mtb genome. The *M. smegmatis perM*-DUC mutant was generated as described (*Schnappinger et al., 2015*). Complementation with *ftsB* was either on a multicopy plasmid or integrated in the chromosomal attL5 site under control of the hsp60 promoter. For FtsB$_{mtb}$ fusion proteins, N-terminal GFP, mCherry, Flag or SNAP tags were introduced to FtsB$_{mtb}$ by PCR. PerM$_{mtb}$ fusion proteins were cloned by fusing mCherry or GFP to the C-terminus of PerM$_{mtb}$. The genotypes of all strains are reported in the key resources table.

## Mouse infections

Female 7–8 week old C57BL/6 mice (Jackson Laboratory) were infected using an inhalation exposure system (Glas-Col) with mid-log phase Mtb culture delivering approximately $10^2$ or $10^3$ bacilli per mouse. At each time point, lungs were homogenized in PBS and bacteria were enumerated by plating serially diluted homogenates on 7H10 agar.

## Replication clock

Experiments were performed as described previously (*Gill et al., 2009*). Mtb strains were transformed with the 'replication clock' plasmid pBP10. For in vitro experiments, cultures were maintained in log phase by sub-culturing every 3 days for ~20 generations in the absence of antibiotics. The plasmid carriage of each Mtb strain was calculated as the number of CFU on 7H10 agar with 30 µg/ml kanamycin divided by the number of CFU on non-selective plates. Segregation rates (s) and errors were calculated using mathematical equations developed by *Gill et al. (2009)* and the error propagation equation. We determined the plasmid segregation constants for WT ($S_{WT} = 0.1904 \pm 0.0516$) and Δ*perM* ($S_{\Delta}=0.1586 \pm 0.0349$) before measuring cell replication rates in vivo.

For in vivo experiments, cell replication rates (r) were calculated as the slopes of the regression lines fitted through plots of ln [f(t)] versus t and multiplied with ($-1/s$), where s is the segregation constant determined in vitro. Death rates were calculated by subtracting slope N from r. Doubling time equals ln (2)/r. The errors of both death rate and doubling time were calculated using error propagation equations. The cumulative bacterial burden (CBB) was calculated using mathematical models developed by *Gill et al. (2009)*.

## In situ Mtb cell length quantification

Mice were infected with log-phase Mtb by aerosol to deliver ~$10^3$ bacilli per mouse. At each time point, lungs were isolated, and the upper left lung lobes were fixed in 10% buffered formalin for 48 hr prior to removal from BSL-3 containment. The lung lobes were subsequently embedded in paraffin, sectioned and stained with acid-fast staining. Slides were blinded before microscopy and the identity of the samples was revealed after length quantifications were completed. Approximately 40 lung sections were examined for each time point with each strain. Tissue slides were visualized in bright-field using Olympus BX60 microscope with a 100x oil objective and images were captured with Olympus DP71 digital camera. Bacteria in the focal plane of the sections were imaged and their cell lengths were quantified using ImageJ (*Schneider et al., 2012*). The remaining lung lobes were homogenized in PBS and cultured on 7H10 agar to determine CFU.

## High-resolution microscopy

Microscopy imaging was performed with the same methods and equipment as previously described (*Botella et al., 2017b*). For length measurements, Mtb or *M. smegmatis* cultures were collected at indicated time points and washed with PBS containing 0.05% Tween 80 (PBST). The Mtb samples were fixed with 4% paraformaldehyde for 4 hr prior to removal from BSL-3 containment. Single cell suspensions were prepared by centrifugation at 800 rpm for 10 min. After spreading on soft agar pads, bacteria were visualized with appropriate filter sets. For peptidoglycan labeling, 1 mM HCC-amino-D-alanine (HADA) or NBD-amino-D-alanine (NADA) was added to *M. smegmatis* cultures as indicated. Aliquots were removed, washed three times with PBST, and then fixed with 4% paraformaldehyde for 30 min before microscopy. The nucleoid staining was performed by incubating *M. smegmatis* cultures with 5 μM SYTO 13 for 5 min. Images were analyzed with ImageJ and Icy (*de Chaumont et al., 2012*; *Schneider et al., 2012*).

For time lapse microscopy experiments, a drop of *M. smegmatis* culture (5 ~ 10 μl) was placed on a glass bottom microwell dish (MatTek, 35 mm, 14 mm microwell), and 1% low melting point agarose (UltraPure, Invitrogen) prepared in 7H9 broth was added gently to the dish until it covered the bacteria drop. Agarose was left to solidify prior to microscopy. Cells were visualized by fluorescence microscopy and snapshots were captured every 15 min. We analyzed the time lapse movies with ImageJ (*Schneider et al., 2012*). First, the fluorescence intensities were measured lengthwise across a single cell at every time point, from which we determined the maximum fluorescence intensities of the middle 1/2 of the cell. Next, each time point was interpolated to 1/10 of the cell cycle of each cell to allow many bacterial cells to be averaged consistently.

## Forward genetic screen

A total of $10^8$ CFU of Msm *perM*-DUC mutant were transformed with an Mtb genomic library generated as described (*Niederweis et al., 1999*). Bacteria were cultured on 7H10 agar with 400 ng/ml atc and 50 μg/ml hygromycin. Colonies were picked and lysed in 20 μl ice-cold 10% glycerol and transformed into 100 μl *E. coli* top10 competent cells by electroporation. The transformants were selected on LB agar containing 200 μg/ml hygromycin. Plasmids were isolated and sequenced to identify Mtb DNA fragments.

## Immunoprecipitation and mass spectrometry

150 ml of Mtb culture in mid-log phase was pelleted and washed twice with PBST. Cells were resuspended in 600 μl lysis buffer (50 mM Tris-HCl, 50 mM NaCl, pH 7.4) containing protease inhibitor cocktail (Roche) and subjected to mechanical lysis with 0.1 zyrconia beads in a homogenizer. After adding 1% dodecyl maltoside, we incubated the protein lysates on ice for 2 hr. Lysates were clarified by centrifugation for 2 min at 7,500 g. 50 μl anti-Flag M2 affinity beads (Sigma, A2220) were added to supernatants and incubated overnight shaking at 4°C. Beads were collected by centrifugation and washed 5 times with PBS. Proteins bound to beads were eluted by incubation with 100 ng/μl flag peptide (Sigma) 1 hr shaking at 4°C. Samples were boiled in Laemmli sample buffer (Bio-Rad) for 10 min prior to removal from BSL-3 containment. Peptides in the eluates were identified by mass spectrometry. Results were analyzed using Scaffold 4 software. Non-specific binding peptides were removed from the results by setting the filter of 'Total Spectrum Count' of each replicate to '<2' in the control samples and '>3' in Flag-FtsB samples. Hits were ranked based on 'Sum Total Spectrum Count' of the Flag-FtsB samples. The 'Sum Total Spectrum Count' is the sum of two independent biological replicates. Hits were assigned utilizing publicly available databases PATRIC (https://www.patricbrc.org/) and FLUTE (http://orca2.tamu.edu/U19/). For pulldown validations, 150 ml M. *smegmatis* was grown to mid-log phase and protein lysates were collected as described above. Protein lysate (10–30 mg) was incubated with 25 μl anti-GFP mAb-agarose beads or 50 μl anti-GFP mAb-magnetic beads (MBL) overnight shaking at 4°C. After incubation, beads were collected by centrifugation or using a DynaMag-Magnet 2 rack (Thermo Fisher scientific) and washed three times with PBS. Proteins bound to beads were eluted by boiling in Laemmli sample buffer for 10 min and analyzed by western blotting.

## Antibodies and western blots

Rabbit polyclonal antibody for FtsB was generated by GenScript. DlaT antibody (*Bryk et al., 2002*) was a gift from R. Bryk and C. Nathan at Weill Cornell Medicine. GFP (Clontech #632380), mCherry (Clontech #632543) and SNAP (NEB #P9310S) antibodies are commercially available. All secondary antibodies were purchased from LI-COR biosciences.

For western blots, protein lysates were prepared by mechanical lysis with 0.1 mm zirconia beads, followed by 2 hr incubation on ice with 1% dodecyl maltoside. Unbroken bacterial cells and beads were removed by centrifugation. Mtb protein lysates were filtered using 0.22 µm spin-X columns (Corning) prior to removal from BSL3 containment. Protein lysates were separated using SDS-PAGE and transferred to a nitrocellulose membrane. For FtsB analysis, membrane was cut at 37kD and incubated with either FtsB antibody or DlaT antibody. After washing and incubation with secondary antibodies, proteins were visualized using Odyssey Infrared Imaging System (LI-COR Biosciences).

## Analysis of protein stability

Chloramphenicol (Sigma) stock was prepared at 100 mg/ml in ethanol and added to mid-log phase Mtb cultures at a final concentration of 20 µg/ml. At each time point, 30 ml Mtb cultures were removed and protein lysates collected as described above. Stability of FtsB was analyzed by western blotting.

## Mtb growth assays

Mtb was grown to mid-log phase in Middlebrook 7H9 medium prior to each experiment. The precultures were pelleted at 4000 rpm for 10 min and washed once in PBS, then diluted to OD 0.05 in assay media. For growth curves in acidic or low iron conditions, Mtb cultures were diluted to OD 0.05 in assay media at day 6. Conditions for the growth curves were as follows: Magnesium-free Sauton's media was prepared as described (*Goodsmith et al., 2015*). Magnesium was supplemented in the form of $MgCl_2$ at the indicated concentrations. 7H9 medium at pH 5.5 or pH 7 were prepared by adjusting 7H9 broth containing 0.2% glycerol, 0.2% dextrose, 0.5% fatty-acid-free BSA (Roche), 0.085% NaCl, 1.95% MES, and 0.05% tyloxapol to pH as indicated. 0.005% sodium oleate was supplemented every other day. Medium with defined iron concentrations was prepared by chelating Sauton's medium (0.4% L-asparagine, 0.27% sodium citrate tribasic dehydrate, 0.015% citric acid, 0.05% potassium phosphate dibasic, and 6% glycerol) with 1% Chelex-100 (sodium form, Bio-Rad) twice for 24 hr. Subsequently Chelex-100 was removed by filtration and 0.05% magnesium sulfate heptahydrate, 5 µM zinc sulfate, 0.1% ethanol, and 0.05% tyloxapol was added to the medium and the medium was adjusted to pH 7.4. Iron was supplemented in the form of $FeCl_3$ at the indicated concentrations.

## Statistical analysis

All statistical analyses were performed using Prism 7 (GraphPad Software) with the statistical tests indicated in figure legends and the corresponding two-tailed P-values reported. *, p<0.05, **, p<0.01, ***, p<0.001 and ****, p<0.0001 were considered significant results.

## Materials and correspondence

Further information and requests for resources and reagents should be directed to the corresponding author Sabine Ehrt (sae2004@med.cornell.edu).

## Acknowledgements

We thank C Nathan, K Rhee (Weill Cornell Medicine) and J Rock (Rockefeller University) for insightful suggestions. We thank J Xavier (Memorial Sloan Kettering Cancer Center) for helpful discussions as well as advice on mathematical calculations. We thank KG Papavinasasundaram (University of Massachusetts) and SA Shaffer (University of Massachusetts Mass Spectrometry Facility) for LC-MS/MS analysis and A North at the Bio-Imaging Resource Center (Rockefeller University) for microscopy training and advice. We thank D Sherman (University of Washington) for providing the pBP10 plasmid, R Bryk and C Nathan (Weill Cornell Medicine) for the DlaT antibody and A Fay (Memorial Sloan Kettering Cancer Center) for advice. We thank P Tamayo, J McConnell and C Healy (Weill Cornell

Medicine) for assisting with the animal experiments. We thank S Schrader (Weill Cornell Medicine) for careful editing of the manuscript.

## Additional information

### Funding

| Funder | Grant reference number | Author |
|---|---|---|
| National Institute of Allergy and Infectious Diseases | U19 AI111143 | Dirk Schnappinger<br>Sabine Ehrt |
| National Institute of Allergy and Infectious Diseases | U19 AI107774 | Dirk Schnappinger<br>Sabine Ehrt |

The funders had no role in study design, data collection and interpretation, or the decision to submit the work for publication.

### Author contributions

Ruojun Wang, Conceptualization, Resources, Formal analysis, Investigation, Visualization, Methodology, Writing—original draft, Writing—review and editing; Kaj Kreutzfeldt, Investigation, Methodology, Writing—review and editing; Helene Botella, Julien Vaubourgeix, Methodology, Writing—review and editing; Dirk Schnappinger, Conceptualization, Funding acquisition, Methodology, Writing—review and editing; Sabine Ehrt, Conceptualization, Resources, Supervision, Funding acquisition, Visualization, Methodology, Writing—original draft, Project administration, Writing—review and editing

### Author ORCIDs

Ruojun Wang (ID) https://orcid.org/0000-0002-0519-2350
Sabine Ehrt (ID) https://orcid.org/0000-0002-7951-2310

### Ethics

Animal experimentation: The animal experiments were performed in accordance with National Institutes of Health guidelines for housing and care of laboratory animals and according to institutional regulations after protocol review and approval by the Institutional Animal Care and Use Committee of Weill Cornell Medicine (Protocol Number 0601441A).

### Decision letter and Author response

Decision letter https://doi.org/10.7554/eLife.49570.sa1
Author response https://doi.org/10.7554/eLife.49570.sa2

## Additional files

### Supplementary files

• Transparent reporting form

### Data availability

All data generated or analyzed during this study are included in the manuscript and supporting files. Source data files have been provided for Figures 1, 2, 3, 4 and 6 and Table 1.

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
