## [Decision Letter]

**Acceptance summary:**

This study demonstrates that a mycobacterial protein, PerM, is required for cell division during chronic tuberculosis infection in mice. Further interrogation of PerM function in *Mycobacterium tuberculosis* and the non-pathogenic model host, *Mycobacterium smegmatis* revealed that PerM is required for the formation of septa in mycobacteria, hence facilitating cell division. The authors’ approach demonstrates that PerM exerts this function through interaction with, and stabilization of, FtsB, a component of the bacterial divisome. This work provides novel insight on mycobacterial division mechanics during tuberculosis infection in an animal model system. The clever use of combinations of fluorescent D-amino acid probes to demonstrate that PerM depletion leads to reduced septum formation demonstrates how these probes can be used to gain novel mechanistic insight in disease-specific model systems. The value of your study is further enhanced through the use of a replication clock plasmid that allows for a coupled view of cell division and DNA replication during in vivo infection, something that is not often seen in the literature. This study also paves the way to application of similar approaches to studying mycobacterial cell division in human disease and the effects of antibiotic therapy on these events. The techniques applied are also relevant for the study of cell division in other bacteria.

**Decision letter after peer review:**

Thank you for submitting your article "Persistent *Mycobacterium tuberculosis* infection in mice requires PerM controlled cell division" for consideration by *eLife*. Your article has been reviewed by three peer reviewers, one of whom is a member of our Board of Reviewing Editors, and the evaluation has been overseen by Wendy Garrett as the Senior Editor. The following individuals involved in review of your submission have agreed to reveal their identity: Cara Boutte (Reviewer #2); E Hesper Rego (Reviewer #3).

The reviewers have discussed the reviews with one another and the Reviewing Editor has drafted this decision to help you prepare a revised submission.

Summary:

Cell division in mycobacteria differs from other well-characterized organisms such as *E. coli* in that elongation occurs at the cell pole in mycobacteria whilst *E. coli* cells elongate by insertion of cell wall material along the sidewall. As a result of this, the divisome machinery is spatially well segregated from the elongasome and upon cell division, divisome components need to disassemble or remodel to allow for development and subsequent extension of the new pole. Considering this, the mycobcacterial divisome requires further characterization to elucidate how these cells coordinate and regulate the division process. Previously, this group identified PerM, an integral membrane protein in *Mycobacterium tuberculosis* that was required for persistence of tubercle bacilli during the chronic phase of infection in the murine model of TB infection. Preliminary evidence suggested that PerM was involved in cell division however, the mechanistic basis for this phenotype was unclear. In this study, the authors attempt to further unravel how PerM is involved in division.

Key findings:

1) Using a replication clock plasmid that is lost if *M. tuberculosis* replicates in mouse lungs, the authors confirm that their previously reported persistence defect for a perM mutant during chronic infection was due to the inability to replicate after the onset of adaptive immunity. The doubling time for the mutant during chronic infection was significantly longer than that of the wild type and this defect was seen with both a high and low initial dose of aerogenic infection. PerM mutant cells were longer than that of the wild type during chronic infection, suggesting that the persistence defect was due to defective cell division

2) They further study this in *M. smegmatis* with a perM knockdown strain, as perM appears to be essential in this case. Depletion of perM using a dual control depletion system, resulted in the formation of long cells that were filamentous with either one or no septum. PerM depletion was bacteriostatic in this case

3) Using a forward genetic screen to identify suppressors for perM essentiality in *M. smegmatis*, the authors identified FtsB, a member of the FtsQLB complex that is implicated in septum synthesis. They confirm that PerM interacts with FtsB using a co-immune precipitation assay with whole cell lysates and localization of fluorescently tagged proteins, both of which localized at the septum. Using time-lapse microscopy, the authors show that PerM localizes to the septum after FtsB.

4) PerM appears to stabilize FtsB as levels of this protein were reduced when perM was deleted in *M. tuberculosis*. Ectopic expression of FtsB rescued the persistence defect in mice.

Essential revisions:

1) Figure 1—figure supplement 1A – Please confirm that this is the correct graph, as there is apparently no error, and it does not look like a typical growth curve. *M. tuberculosis* should double every cell cycle, it doesn't increase by 10-fold, so the growth curve shouldn't be linear on a log10 axis. Unless there was no lag? What does continuous culture mean in this context? Are OD values really as high as 10e6?

2) It's hard to see the reduced colony sizes alluded to in Figure 3A – median colony diameters should be provided

3) The authors indicate that PerM localized to mid-cell 15 minutes after FtsB, this is not clear from the data they provide, which are images in some cases without time stamps. A better way of illustrating this is to plot the localization of PerM and FtsZ as a function of time between division cycles.

4) Some of the most important mechanistic data in this paper is the protein stability data in Figures 4B and D. However, this does not appear to have been replicated. Please show quantification of these gel bands, with error. The results need to be shown to be reproducible.

5) The conclusion that PerM is involved in septum formation is somewhat confusing. If that were true, it seems that the elongated cells would have fewer septa. Instead, they have more (Figure 2C) than WT, suggesting a defect in late septum formation/cell separation. Indeed, the perM depletion mutant does not truly phenocopy a ftsB depletion which forms long, branching cells. Please re-assess the conclusion that PerM is involved in septal formation or address the discrepancy in perM depletion vs. ftsB depletion phenotypes.

---

## [Author Response]

Essential revisions:1) Figure 1—figure supplement 1A – Please confirm that this is the correct graph, as there is apparently no error, and it does not look like a typical growth curve. M. tuberculosis should double every cell cycle, it doesn't increase by 10-fold, so the growth curve shouldn't be linear on a log10 axis. Unless there was no lag? What does continuous culture mean in this context? Are OD values really as high as 10e6?

We apologize that the figure legend was confusing. For this experiment, *M. tuberculosis* cultures were maintained in log phase by sub-culturing every three days for ~20 generations. We started the culture at OD_600_ = 0.05 and diluted it to OD_600_ = 0.05-0.1 every three days. We have replotted the data and edited the figure legend to more accurately describe the experiment.

2) It's hard to see the reduced colony sizes alluded to in Figure 3A – median colony diameters should be provided

We agree and have removed comments on colony sizes from the main text. We included the incubation time for each plate in the figure legend to allow for judgement of relative growth.

3) The authors indicate that PerM localized to mid-cell 15 minutes after FtsB, this is not clear from the data they provide, which are images in some cases without time stamps. A better way of illustrating this is to plot the localization of PerM and FtsZ as a function of time between division cycles.

We have quantified the time lapse videos, as suggested, and included the data in Figure 5B in the revised manuscript.

4) Some of the most important mechanistic data in this paper is the protein stability data in Figures 4B and D. However, this does not appear to have been replicated. Please show quantification of these gel bands, with error. The results need to be shown to be reproducible.

We have included the biological replicates of both figures and quantifications of band intensities as Figure 5—figure supplement 3 in the revised manuscript.

5) The conclusion that PerM is involved in septum formation is somewhat confusing. If that were true, it seems that the elongated cells would have fewer septa. Instead, they have more (Figure 2C) than WT, suggesting a defect in late septum formation/cell separation. Indeed, the perM depletion mutant does not truly phenocopy a ftsB depletion which forms long, branching cells. Please re-assess the conclusion that PerM is involved in septal formation or address the discrepancy in perM depletion vs. ftsB depletion phenotypes.

We have performed pulse-chase experiments with two fluorescent D-alanine probes to address this concern (new Figure 3). Our data demonstrated that PerM depletion led to impaired septum formation and separation in *M. smegmatis*. This result agrees with the observation in Figure 2 that a higher fraction of PerM-depleted bacteria than WT contained one septum.

It is true that PerM depletion did not fully phenocopy FtsB-depleted cells. This may be due to continued FtsB synthesis in the *perM*-DUC strain, and that reducing FtsB stability following PerM depletion may not reduce FtsB levels to the same extent as repressing FtsB expression in the FtsB-depletion strain.